# Tailings-flow runout analysis: Examining the applicability of a semi-physical area-volume relationship using a novel database

Negar Ghahramani[1], Andrew Mitchell[1], Nahyan M. Rana[2], Scott McDougall[1], Stephen G. Evans[2], Andy Take[3]

[1]Department of Earth, Ocean and Atmospheric Sciences, The University of British Columbia, Vancouver, V6T 1Z4, Canada
[2]Department of Earth and Environmental Sciences, University of Waterloo, Waterloo, N2L 3G1, Canada
[3]Department of Civil Engineering, Queen's University, Kingston, K7L 3N6, Canada

*Correspondence to*: Negar Ghahramani (nghahramani@eoas.ubc.ca )

**Abstract.**

Tailings-flows result from the breach of tailings dams. Large-scale tailings-flows can travel over substantial distances with high velocities and cause significant life loss, environmental damage and economic costs. Runout modelling and inundation mapping are critical components of risk assessment for tailings dams. In an attempt to develop consistency in reporting tailings data, we established a new tailings-flow runout classification system. Our data analysis applies to the zone corresponding to the extent of the main solid tailings deposit, which is characterized by visible or field-confirmed sedimentation, above typical surface water levels if extending into downstream water bodies. We introduced a new database of 33 tailings dam breaches by independently estimating the planimetric inundation area for each event using remote sensing data. This paper examines the applicability of a semi-physical area-volume relationship using the new database. Our results indicate that the equation $A=cV^{2/3}$, which has been used previously to characterize the mobility of other types of mass movements, is a statistically-justifiable choice for the relationship between total released volume and planimetric inundation area. Our analysis suggests that, for a given volume, tailings-flows are, on average, less mobile than lahars but more mobile than non-volcanic debris flows, rock avalanches and waste dump failures.

## 1 Introduction

Tailings dams are a critical piece of mining infrastructure (Blight, 2009). These dams retain mine tailings, a waste product of the mineral processing operations that includes finely ground rock and process water. Some of these wastes may classify as hazardous material (Vick, 1990). When a tailings dam breach occurs, a destructive flow of mine tailings can develop (e.g. (Macías et al., 2015)). These flows may travel over substantial distances and impact large areas (Rico et al., 2008a). The ability to understand and predict the motion of flowing tailings is a crucial step in protecting people, infrastructure and the environment from these events.

More than 350 tailings dam breaches have been recorded worldwide since the early twentieth century (Chambers and Bowker, 2019; International Commission on Large Dams (ICOLD), 2001; Rico et al., 2008a; Santamarina et al., 2019; WISE, 2020) (Fig. 1). The records indicate that approximately one-third of these events led to loss of life and/or the release of more than 100,000 $m^3$ of tailings and/or water (Chambers and Bowker, 2019). For example, the fluorite tailings dam at Stava, Italy failed in 1985 and released a total volume of 185,000 $m^3$ of muddy debris. As a result, the Stava and Tesero villages were destroyed

and 243 people lost their lives (Chandler and Tosatti, 1995; Luino and De Graff, 2012; Pirulli et al., 2017; WISE, 2020). The 2014 Mount Polley tailings dam failure in British Columbia, Canada resulted in the release of about 25 million cubic meters (M $m^3$) of water and tailings into Polley Lake, Hazeltine Creek, and Quesnel Lake. The tailings inundation area was estimated to be approximately 2.4 M $m^2$ (Cuervo et al., 2017; Mount Polley Mining Corporation, 2016). The 2015 Fundão tailings dam failure in Brazil resulted in the release of about 35 M $m^3$ of tailings materials. This event killed 19 people and caused long-

lasting environmental damage to several water channels in the basin of the Doce River (Carmo et al., 2017; Hatje et al., 2017; WISE, 2020). More recently, another disastrous tailings dam breach occurred at the Feijão mine near Brumadinho, Brazil in January 25, 2019. Almost 12 M $m^3$ of tailings left the impoundment and the resulting tailings-flow travelled for almost 9 km and inundated an area of approximately 3.0 M $m^2$ before reaching the Rio Paraopeba (WISE, 2020); 259 people were reported killed, and 11 were reported missing as a result of this failure (WISE, 2020). All of these events also resulted in substantial

financial losses for the mine operators and investors.

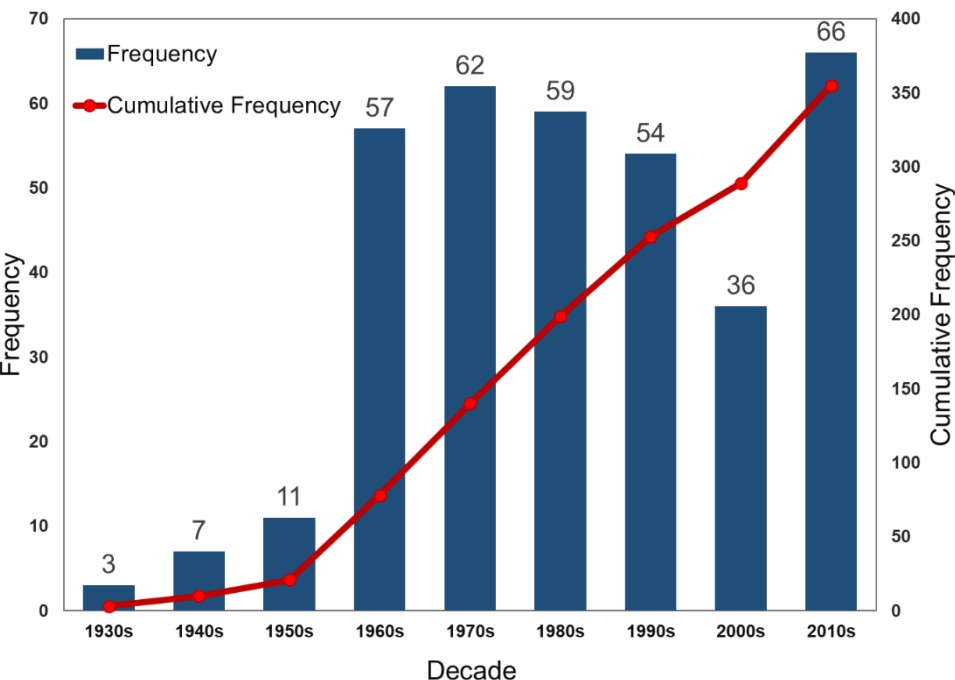

**Figure 1: Decadal frequency and cumulative frequency of tailings dam breaches worldwide (n= 355) between 1930 and 2019. Sources: (Chambers and Bowker, 2019; ICOLD, 2001; WISE, 2020)**

Tailings dam breach runout analysis studies the behaviour of tailings-flows. The term "tailings-flow" refers to various forms of tailings outflow movement resulting from the breach of a tailings dam. This may include a partial or a total release of the stored tailings and associated water (Blight, 2009; Rico et al., 2008a, 2008b; Villavicencio et al., 2014). Tailings-flows exhibit different characteristics depending on various factors, including sediment concentration, the presence of surface water, embankment configuration, failure mechanism, liquefaction potential, and downstream topography (Martin et al., 2019; Small et al., 2017). Tailings-flows can take various forms, ranging from a massive debris flood consisting of water and sediment, to a flowslide (Hungr et al., 2014). These flows can travel long distances at extremely rapid velocities (> 5 m/s) (Blight, 1997; Blight et al., 1981; Harder and Stewart, 1996; Jeyapalan et al., 1983a, 1983b; Kossoff et al., 2014; Macías et al., 2015; Rico et al., 2008a). Runout modelling and inundation mapping of tailings dam breaches are essential steps for estimating the potential consequences of a tailings dam failure, determining appropriately stringent design criteria, and developing emergency response and preparedness plans (Canadian Dam Association (CDA), 2014; Knight Piésold, 2014; Martin et al., 2015, 2019; McDougall, 2017). In recent years, there has been an increase in the study of the consequences of tailings dam breaches following several major disasters worldwide (Roche et al., 2017; Santamarina et al., 2019; Schoenberger, 2016). However, much uncertainty still exists in this field. The number of available empirical-statistical runout models is limited (Section 2). Most of the commonly used numerical models were developed primarily for either clear water flood analysis (Brunner, 2016; Danish Hydraulic Institute (DHI), 2007; Martin et al., 2019) or the analysis of flow-like landslides (McDougall, 2017; McDougall and Hungr, 2004; Pastor et al., 2002; Pirulli et al., 2017) and do not necessarily account for the compositional variety of tailings and its potential influence on the downstream flow behaviour (Dibike et al., 2018; Macías et al., 2015; Pirulli et al., 2017). Due to these limitations, hazard maps delimiting potential inundation areas based on current techniques may not reliably characterize the extent and intensity (e.g. flow depth and velocity) of possible tailings dam breach scenarios.

The purpose of this paper is to i) introduce a runout zone classification method in an attempt to develop consistency in reporting runout distances and inundation areas of tailings-flows, ii) introduce a new database of 33 tailings dam breaches where released volume was reported and the planimetric inundation areas were quantitatively measured for all of the events, iii) examine the applicability of a semi-physical area-volume relationship for tailings-flow cases to help characterize the mobility and potential impacts of these types of failures, and iv) investigate the effects of additional attributes of the tailings and downstream topography, such as tailings mine types and confinement of travel path, which could potentially be used to refine these empirical-statistical relationships. The present work builds on previous work described in (Ghahramani et al., 2019).

## 2 Previous Empirical Studies

### 2.1 Empirical runout studies for tailings dam breaches

Empirical runout analysis of tailings dam breaches is a relatively new research topic. Rico et al. (2008a) proposed a set of empirical correlations that relate tailings-flow characteristics (e.g. released volume and runout distance) to the geometric

characteristics of tailings dams (e.g. dam height and total impoundment volume). A database of 28 tailings dam breaches (from 1965 to 2000) containing information on released volume and runout distance was used in their study (Rico et al., 2008a).

Rico et al. (2008a) found positive correlations between i) the total volume of the tailings in the impoundment at the time of failure and the tailings released volume, and ii) the tailings released volume and the tailings runout distance. The tailings released volumes in their work were extracted from existing databases and publications with no information on the distinction between the volume of released solid tailings, interstitial (pore) water, and surface (free) water. A parameter referred to as the "dam factor" (the product of the dam height and tailings released volume, $H \times V_F$) was used to improve the correlations in their study. This parameter was originally developed by Hagen and the Committee on the Safety of Existing Dams for the peak discharge estimation of water dam-breaks (Committee on the Safety of Existing Dams, 1983; Costa, 1985; Hagen, 1982). The related equations by Rico et al. (2008a) are provided in Table 1.

The Tailings Dam Breach Working Group (WG) of the Canadian Dam Association (CDA) Mining Dam Committee compiled a tailings dam breach database that includes the 28 cases presented by Rico et al. (2008a) (Rico et al., 2008a)(Rico et al., 2008a)(Rico et al., 2008a)(Rico et al., 2008a)(Rico et al., 2008a)(Rico et al., 2008a)(Rico et al., 2008a)(Rico et al., 2008a), plus 51 additional cases (Small et al., 2017). Their study discussed the limited information provided in the Rico et al. (2008a) database and listed additional factors that could influence the behaviour of tailings flows. The WG proposed a four-element classification matrix based on two main factors: i) the presence of free standing water in close proximity to the breach, and ii) tailings liquefaction potential. The empirical relationships of Rico et al. (2008a), were re-examined based on the proposed classification (Small et al., 2017).

Larrauri and Lall (2018) updated the database presented in Rico et al. (2008a) and reexamined their empirical correlations. They introduced a new predictor, called $H_f$ which is defined as $H \times (V_F / V_T) \times V_F$, where $V_T$ is the total volume of the tailings impoundment and $V_F$ is the total released volume. Using the updated database, they concluded that the relationship between $H_f$ and runout distance has a stronger correlation ($R^2 = 0.53$, Table 1) than the relationship between dam factor and runout distance ($R^2 = 0.44$) (Larrauri and Lall, 2018). However, arguably both correlations are fairly weak and the physical basis of the $H_f$ factor was not discussed in their study. Rico et al. (2008a) and Larrauri and Lall (2018) both noted that uncertainties in their databases suggest that the results need to be treated with caution.

**Table 1: Empirical relationships proposed by others for the runout analysis of tailings dam breaches**

| Input Parameter | Output Parameter | Equation | R-squared | References |
|---|---|---|---|---|
| Impoundment Volume ($V_T$) | Total Released Volume | $V_F = 0.354V_T^{1.01}$ | 0.86 | (Rico et al., 2008a) |
| Total Released Volume ($V_F$) | Maximum Runout Distance | $D_{max} = 14.45V_F^{0.76}$ | 0.56 | (Rico et al., 2008a) |
| Dam Height ($H$) | Maximum Runout Distance | $D_{max} = 0.05H^{1.41}$ | 0.16 | (Rico et al., 2008a) |
| Dam Factor ($HV_F$) | Maximum Runout Distance | $D_{max} = 1.61(HV_F)^{0.66}$ | 0.57 | (Rico et al., 2008a) |
| Impoundment Volume ($V_T$) | Total Released Volume | $V_F = 0.332V_T^{0.95}$ | 0.89[a] | (Larrauri and Lall, 2018) |
| $H_f$ ($H(V_F/V_T)V_F$) | Maximum Runout Distance | $D_{max} = 3.04H_f^{0.545}$ | 0.53[a] | (Larrauri and Lall, 2018) |

[a]These values were calculated using a different database from Rico et al. (Rico et al., 2008a). In all the relationships, $V_T$ and $V_F$ are in $10^6$ m$^3$, $H$ is in m, and $D_{max}$ is in km.

## 2.2 Empirical runout relationships - area and volume

Several authors have investigated the relationship between inundation or deposit area ($A$) and flow volume ($V$) for different
types of flow-type landslides (e.g. (Berti and Simoni, 2007; Davies, 1982; Delaney and Evans, 2014; Golder Associates Ltd.,
1995; Griswold and Iverson, 2008; Hungr, 1981; Hungr and Evans, 1993; Iverson et al., 1998; Li, 1983; Simoni et al., 2011))
(Table 2). Li (1983) presented an empirical relationship between rock avalanche deposit area and volume for 76 major
European rock avalanches. The deposit area and volume were logarithmically transformed to apply a linear least-squares
regression analysis (Li, 1983). Hungr and Evans (1993) applied a similar methodology to a different dataset of rock avalanches.
However, they made an assumption that the deposits at various scales retain a similar geometry, which resulted in the following
scaling relation for the area-volume relationship:

$$A = cV^{2/3} \tag{1}$$

where $A$ is the inundation area, $V$ is the total flow volume and $c$ is a constant related to flow mobility (Hungr and Evans, 1993)
(i.e. for a given event volume, a higher mobility flow results in a higher planimetric inundation area). Golder Associates Ltd.
(1995) derived an area-volume relationship for mine waste dump failures using a database of 22 cases. Iverson et al. (1998)
presented similar area-volume relationships as in Hungr and Evans (1993) for lahars (Table 2). Statistical analysis of a dataset
containing 27 lahars was used to calibrate and test the validity of those equations (Iverson et al., 1998). Berti and Simoni
(2007) and Griswold and Iverson (2008) extended the same methodology to non-volcanic debris flows. Griswold and Iverson
(2008) also substantially expanded the database of rock avalanches and found a slightly different correlation than Hungr and
Evans (1993) (Table 2).

**Table 2: Selected empirical relationships between volume and inundation area proposed by others for various landslide types**

| Database Type | Equation | n | R-squared | c Coefficient in Eq. 1 | References |
|---|---|---|---|---|---|
| Rock avalanches | $A = 76\ V^{0.57}$ | 76 | 0.78 | - | (Li, 1983)[a] |
| Rock avalanches | $A = 12\ V^{2/3}$ | 40 | - | 12 | (Hungr and Evans, 1993)[b] |
| Lahars | $A = 200\ V^{2/3}$ | 27 | 0.90 | 200 | (Iverson et al., 1998)[b] |
| Debris flows | $A = 20\ V^{2/3}$ | 44 | 0.91 | 20 | (Griswold and Iverson, 2008)[b] |
| Debris flows | $A = 18\ V^{2/3}$ | 115 | | 18 | (Simoni et al., 2011)[b] |
| Rock avalanches | $A = 20\ V^{2/3}$ | 142 | 0.79 | 20 | (Griswold and Iverson, 2008)[b] |

[a]The original equation from (Li, 1983) is presented in power law format to facilitate comparison.
[b]$A$ and $V$ are planimetric area and flow volume, respectively ($A$ is in m$^2$ and $V$ is in m$^3$).

## 3 Methodology

### 3.1 Dataset compilation

Tailings dam breaches have been recorded since the beginning of the twentieth century (Chambers and Bowker, 2019; ICOLD, 2001). Several compilations and summaries of the characteristics of significant tailings dam breaches can be found in the literature (Chambers and Bowker, 2019; ICOLD, 2001; Small et al., 2017; WISE, 2020). These summaries contain key information about the events, such as dates, causes and triggers of failure, dam heights and construction methods, and the volumes of released and impounded tailings. However, most of the records lack consistency in the reported data related to runout, including information related to factors that may better characterize tailings-flows, due to the lack of a systematic methodology in reporting. In the present study, we first compiled available information for 71 tailings dam breaches and then assessed the runout characteristics of each case individually. Data sources included existing literature on individual tailings dam breach events, existing databases, and remote sensing data obtained from satellite images or aerial photos.

We classified the inundation areas into two zones (Fig. 2). Zone 1 is the primary impact zone, defined as the extent of the main solid tailings deposit, which is characterized by remotely visible or field-confirmed sedimentation, above typical bankfull elevations if extending into downstream river channels. Zone 2 is the secondary impact zone, defined as the area downstream of Zone 1 that is further impacted by the tailings-flow in some form. Secondary impacts may include flood or displacement wave impacts (i.e. fluid impacts above typical downstream water levels) and sediment plume impacts (i.e. below typical downstream water levels).

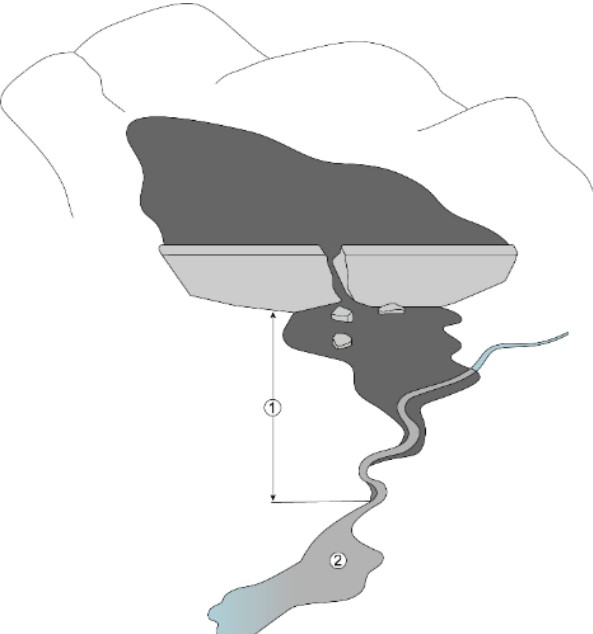

**Figure 2: An idealized representation of a tailings dam breach showing the two runout limit classifications. Zone 1 represents the primary impact zone, defined as the extent of the main solid tailings deposit, which is characterized by remotely visible or field-confirmed sedimentation, above typical water levels if extending into downstream streams. Zone 2 is the secondary impact zone,**

**defined as the area downstream of Zone 1 that is still impacted by the tailings-flow in some form and includes the distal limit of the flow.**

Figure 3 shows a flowchart that summarizes our data compilation methodology, including the screening of data sources, the impact zone classification, the delineation of Zone 1, and the estimation of uncertainty due to image resolution. The extent of Zone 2 is typically more challenging to estimate than the extent of Zone 1, due to the variability of downstream flow mixing

conditions, the relatively transient nature of secondary impacts, and the inherent limitations (e.g. image resolution) of the remote detection methods used. The focus of this study was therefore on Zone 1.

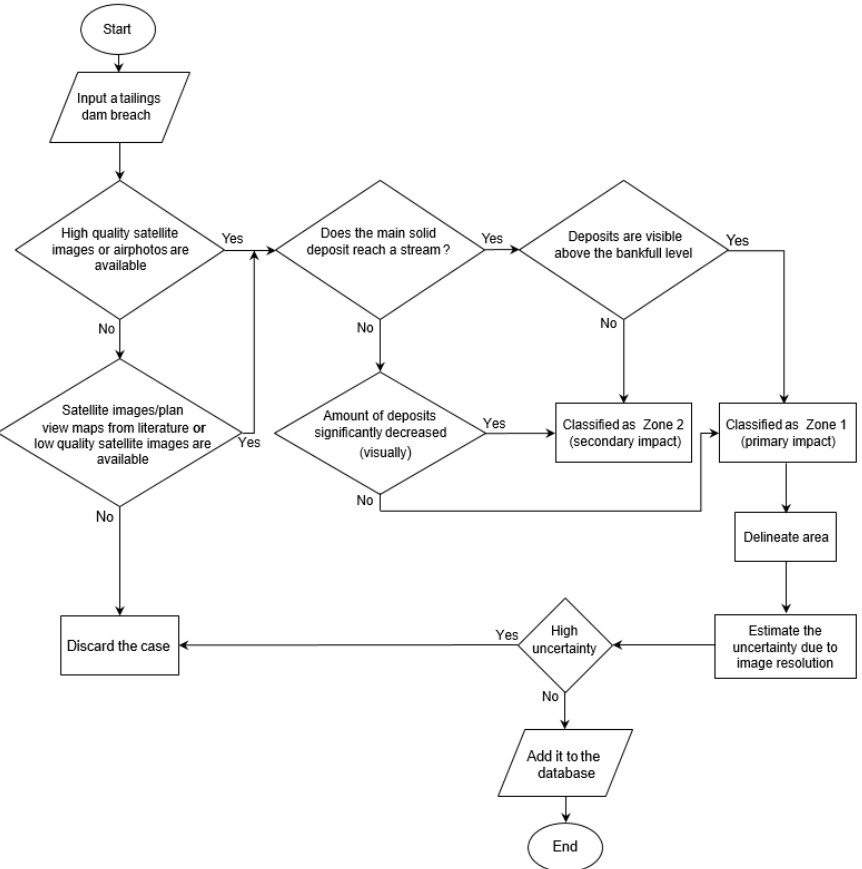

**Figure 3: Schematic representation of the methodology applied to obtain data for tailings-flows inundation area.**

Applying our methodology to the preliminary database comprising 71 tailings dam breaches resulted in 33 cases for which we

were able to obtain satisfactory imagery and independently estimate runout distance and planimetric inundation area (Table 3). Figures 4 and 5 illustrate two examples of delineating the extent of Zones 1 and 2 for the tailings dam breaches at the Feijão mine near Brumadinho, Brazil, 2019, and the Cieneguita mine in Mexico, 2018, respectively. The primary impact zone for Feijão (red-dashed polygon in Fig. 4) was established through a detailed comparison of pre-event and post-event PlanetScope (3 m) imagery. After entering the Paraopeba River, the Feijão tailings-flow exhibited no visible sedimentation above the

bankfull level (blue dashed-line in Fig. 4) and the channel width stayed the same. However, we observed changes in water
colour for over 100 km down-river, which we interpret to represent the secondary impact zone (Zone 2). A similar methodology
was applied for the Cieneguita mine tailings dam breach on June 4, 2018 in Mexico, for which the runout distance was reported
to be between 26 and 29 km (Chambers and Bowker, 2019; WISE, 2020). Based on our methodology, the transition between
Zone 1 and Zone 2 occurs where the extent of the tailings deposits significantly decreased. Normalized Difference Vegetation

Index (NDVI) change detection analysis (Fig. 5 inset (a)) was used to help identify the tailings deposits. The estimated Zone
1 runout distance was approximately 15 km.

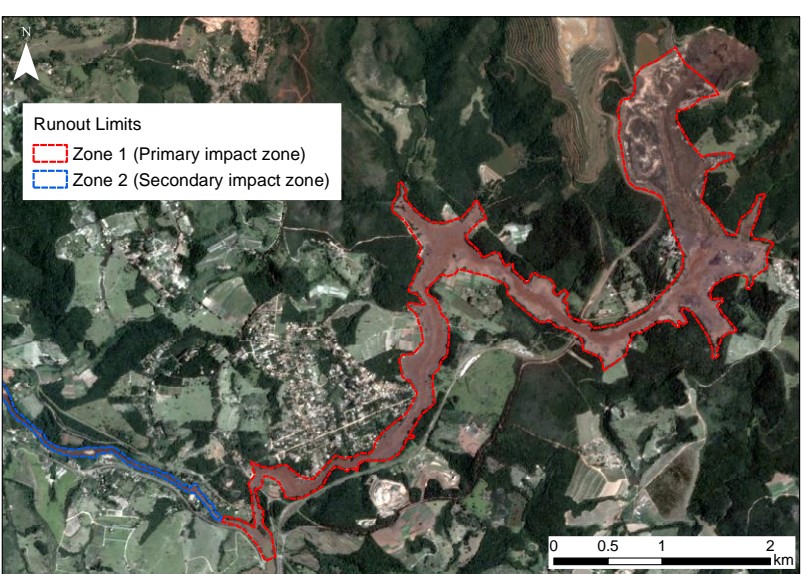

**Figure 4: Aerial view of the tailings dam breach at the Feijao mine near Brumadinho, Brazil, January 25, 2019. Zone 1 is shown in the red dashed polygon. The portion of Zone 2 that is visible in this image is shown in the blue dashed polygon. Image courtesy of**
**Planet Labs, Inc. (January 29, 2019).**

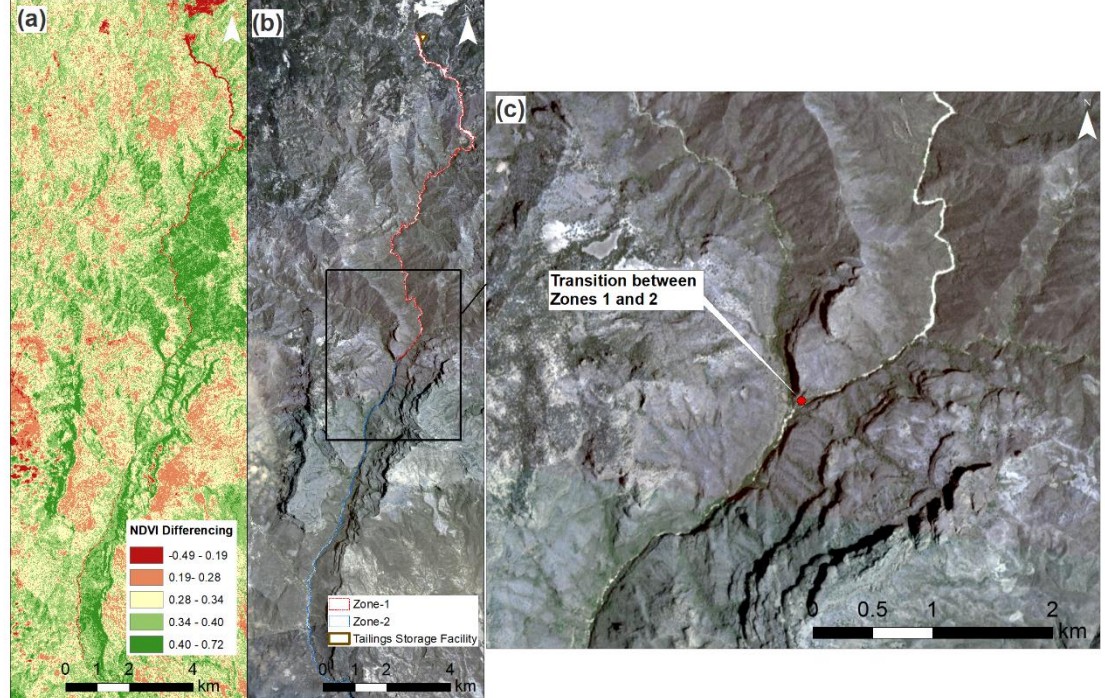

**Figure 5: Aerial views of the tailings dam breach at the Cieneguita mine in Mexico, June 4, 2018. The NDVI differencing change detection technique was used to help delineating Zones 1 and 2 inundation areas (a). Zones 1 and 2 are shown in the red and blue dashed polygons, respectively (b). The inset image (c) shows the transition between Zones 1 and 2 (red dot). Image courtesy of Planet Labs, Inc. (June 12, 2018).**

Compared with the hundreds of tailings dam breach cases listed in previous databases (Chambers and Bowker, 2019; ICOLD, 2001; Rico et al., 2008a; Small et al., 2017), the relatively limited number of cases (33) in our new database reflects the limited availability of suitable imagery, especially for older cases that predate satellite imagery. We used a simple approach to quantitatively estimate the uncertainty due to limitations in image resolution in our area measurements based on the pixel sizes of the images. The maximum percentage uncertainty due to image resolution was considered to be equal to the ratio of the total area of the pixels intersected by the perimeter of Zone 1 to the inundation area multiplied by one hundred. Our database contains information on the percentage uncertainty of each case (Table 3).

Additional key attributes are included in our database (Table 3). We classified our cases using the four elements classification matrix in Small et al. (2017), described above. We also used the following two categories proposed by Golder Associates Ltd. (1995) to classify the confinement of the travel path: i) confined, in which the flow path is constrained by relatively steep side slopes of a gully or valley; and ii) unconfined, in which the flow path is on an open slope or relatively flat surface and the topography permits spreading of the tailings-flow from an early stage. Similarly, to classify the tailings mine type, we used the following two categories introduced by Small et al. (2017): i) hard rock mine tailings, which includes lead-zinc, copper, gold-silver, molybdenum, nickel from sulphide deposits, and uranium; and ii) soft rock mine tailings, which includes coal, potash, fluorite, gypsum, and aluminum (Bussière, 2007; Small et al., 2017; Vick, 1990).

The dam height and total released volume data were collected from existing databases and publications. We also included information on the volume of free water released, if available. However, for the empirical analysis, only the total reported released tailings volume was considered. We note that there is limited information available on how the reported released volumes within the existing databases were obtained (including the distinction between the volume of released solid tailings, interstitial water, and surface water).

**TABLE 3. Database of 33 tailings dam breaches (tailings-flows) containing independently estimated measurements of Zone 1 runout distance and planimetric inundation area.**

| ID | Event | Location | Latitude | Longitude | Date | Confined / Unconfined | Type of Tailings | WG Classification | Tailings Released Volume (M m³) | Total Free Water Released (M m³) | Total Released Volume (M m³) | Zone 1- Tailings Runout Distance (m) | Zone 1- Inundation Area (m²) | Uncertainty Level % |
|---|---|---|---|---|---|---|---|---|---|---|---|---|---|---|
| 1 | Bellavista | Chile | 32.69° S | 70.8° W | 1965-03-28 | Unconfined | Hard | IA | | | 0.055 | 1,300 | 130,000 | 4 |
| 2 | Cerro Negro | Chile | 32.58° S | 70.88° W | 1965-03-28 | Unconfined | Hard | IA | | | 0.07 | 3,200 | 1,300,000 | 1 |
| 3 | El Cobre (New & Old) | Chile | 32.65° S | 71.14° W | 1965-03-28 | Confined | Hard | IA | 0.36 | 2.04 | 2.4 | 11,200 | 5,900,000 | 1 |
| 4 | Los Maquis | Chile | 32.463° S | 71.08° W | 1965-03-28 | Confined | Hard | IA | | | 0.021 | 1,500 | 47,000 | 12 |
| 5 | Sgorigrad | Bulgaria | 43.16° N | 23.51° E | 1966-05-01 | Confined | Hard | IA | | | 0.22 | 6,000 | 400,000 | 5 |
| 6 | Certej | Romania | 45.96° N | 22.98° E | 1971-10-30 | Confined | Hard | IA | | | 0.3 | 2,300 | 380,000 | 3 |
| 7 | Bafokeng | South Africa | 25.52° S | 27.20° E | 1974-11-11 | Confined | Hard | IA | | | 3 | 22,000 | 9,000,000 | 17 |
| 8 | Stava | Italy | 46.32° N | 11.50° E | 1985-07-19 | Confined | Soft | IA | 0.17 | 0.02 | 0.19 | 4,200 | 500,000 | 18 |
| 9 | Stancil | USA | 39.57° N | 76.03° W | 1989-08-25 | Unconfined | Hard | IB | 0.038 | - | 0.038 | 100 | 7,000 | 5 |
| 10 | Tapo Canyon | USA | 34.33° N | 118.72° W | 1994-01-17 | Confined | Hard | 2A | | | 0.055 | 730 | 30,000 | 18 |
| 11 | Merriespruit (Harmony) | South Africa | 28.13° S | 26.85° E | 1994-02-22 | Unconfined | Hard | IA | 0.51 | 0.09 | 0.6 | 2,200 | 900,000 | 19 |
| 12 | Pinto Valley | USA | 33.41° N | 110.96° W | 1997-10-22 | Confined | Hard | 2A | 0.23 | - | 0.23 | 830 | 80,000 | 3 |
| 13 | Los Frailes/ Aznalcollar | Spain | 37.52° N | 6.23° W | 1998-04-24 | Unconfined | Hard | IA | 1.5 | 5.5 | 7 | 29,000 | 16,000,000 | 11 |
| 14 | Comurhex, Cogéma/Areva | France | 43.21° N | 2.98° E | 2004-03-20 | Unconfined | Hard | IA/1B | | | 0.03 | 700 | 70,000 | 2 |
| 15 | Mineracao (Rio Pomba) | Brazil | 21.22° S | 42.68° W | 2007-01-10 | Confined | Soft | IA/1B | | | 2 | 40,000 | 8,000,000 | 38 |
| 16 | Xiangfen | China | 35.88° N | 111.58° E | 2008-09-08 | Unconfined | Hard | IA/1B | | | 0.19 | 2,300 | 400,000 | 7 |
| 17 | Kingston fossil plant | USA | 35.9° N | 84.52° W | 2008-12-22 | Unconfined | Soft | 2A | | | 4.1 | 1,400 | 800,000 | 7 |
| 18 | Karamken | Russia | 60.24° N | 151.06° E | 2009-08-29 | Confined | Hard | IA | 1.2 | 1 | 2.2 | 2,900 | 520,000 | 7 |
| 19 | Las Palmas | Chile | 35.18° S | 71.76° W | 2010-02-27 | Unconfined | Hard | 2A | 0.1 | - | 0.1 | 550 | 80,000 | 3 |
| 20 | Ajka | Hungary | 47.09° N | 17.50° E | 2010-10-04 | Confined | Soft | IA | | | 1.6 | 17,800 | 6,000,000 | 1 |
| 21 | Kayakari | Japan | 38.81° N | 141.53° E | 2011-03-11 | Confined | Hard | 2A | 0.041 | - | 0.041 | 2,000 | 150,000 | 3 |
| 22 | Gullbridge | Canada | 49.2° N | 56.17° W | 2012-12-17 | Unconfined | Hard | IB | 0.1 | 0.0005 | 0.1005 | 500 | 44,000 | 1 |
| 23 | Obed Mountain | Canada | 53.57° N | 117.52° W | 2013-10-31 | Confined | Soft | IB | 0.67 | - | 0.67 | 5,100 | 1,000,000 | 6 |
| 24 | Mount Polley | Canada | 52.51° N | 121.6° W | 2014-08-04 | Confined | Hard | IB | 7.3 | 17.1 | 25.6 | 9,000 | 2,000,000 | 6 |
| 25 | Fundão | Brazil | 20.21° S | 43.47° W | 2015-11-05 | Confined | Hard | 2A | | | 33 | 99,000 | 21,000,000 | 3 |
| 26 | Luoyang | China | 34.7° N | 112.06° E | 2016-08-08 | Confined | Soft | 2A | 2 | - | 2 | 2,500 | 300,000 | 6 |
| 27 | Tonghshan | China | 30.08° N | 114.95° E | 2017-03-12 | Unconfined | Hard | 2A/2B | | | 0.5 | 500 | 300,000 | 5 |
| 28 | Mishor Rotem | Israel | 31.06° N | 35.21° E | 2017-06-30 | Confined | Soft | IB | | | 0.1 | 28,000 | 2,000,000 | 18 |
| 29 | Jharsuguda (Vedanta) | India | 21.78° N | 84.08° E | 2017-08-28 | Unconfined | Soft | 2A/2B | | | 2.6 | 640 | 500,000 | 3 |
| 30 | Cieneguita | Mexico | 27.12° N | 108.03° W | 2018-06-04 | Confined | Hard | Unknown | | | 0.44 | 15,000 | 500,000 | 17 |
| 31 | Cadia | Australia | 33.5° S | 148.99° E | 2018-03-09 | Unconfined | Hard | 2A | 1.33 | - | 1.33 | 480 | 120,000 | 5 |
| 32 | Feijão | Brazil | 20.12° S | 44.12° W | 2019-01-25 | Confined | Hard | 2A | 9.65 | - | 9.65 | 9,000 | 2,700,000 | 3 |
| 33 | Cobriza | Peru | 12.58° S | 74.37° W | 2019-07-10 | Confined | Hard | Unknown | | | 0.07 | 450 | 70,000 | 16 |

[a]The procedures used to classify the cases based on path confinement and tailings type and WG classification matrix can be found in Section 3.1. [b]Information on released volumes was collected from other databases (tailings released volume is the released volume of solids and interstitial water; free water released volume is the released volume of surface water).

## 3.2 Statistical analysis

### 3.2.1 Volume dependency of Zone 1 inundation area

In this study, the scaling relationship adopted in previous studies (Davies, 1982; Golder Associates Ltd., 1995; Griswold and Iverson, 2008; Hungr and Evans, 1993; Iverson et al., 1998; Li, 1983) was applied to the new tailings-flow database. The analysis relates the estimated Zone 1 inundation area (dependent variable) to the reported total released volume (independent variable) in Table 3. A simplifying assumption was made that the released volume approximately matches the volume deposited downstream in Zone 1 (i.e. the potential contribution of entrainment and erosion to the total volume of the deposited material was not considered).

We used our tailings dam breach database (n = 33) to fit a regression model and examine the applicability of Eq. (1) for tailings-flows. We transformed the data into a log-log scale and applied the standard least-squares linear regression method. A linear regression model was fit to the data using a specified 2/3 slope and was compared to the standard least-squares linear regression. The uncertainty in the tailings release volume estimates is not considered for this analysis.

### 3.2.2 Effect of other factors on Zone 1 inundation area

Exploratory analyses were completed to investigate the effects of qualitative factors, such as the tailings mine type and travel path topographic confinement, on the area-volume relationship. This analysis was achieved by creating box plots of the regression residuals and colour-coding the data points in the area-volume plot to visually assess if there were trends that could potentially be incorporated into the regression analysis to reduce the uncertainty.

## 4 Results

Figure 6 shows the log-linear regression line for Zone 1 inundation area as a function of total released volume with the 95% confidence interval of the best-fit regression line. Please note that the 95% confidence intervals account for the uncertainty of the regression line and not the individual observations. The regression with a specified 2/3 slope (i.e. based on Eq. (1)) plots within the 95% confidence interval of the best-fit regression, supporting the hypothesis that this scaling relationship is valid for the tailings breach data. Table 4 compares the output from the regression analysis for the best-fit and the specified 2/3 slope regression models. The following regression equation was obtained in power-law form for the specified 2/3 slope regression model:

$$A = 80 \, V_R{}^{2/3} \tag{2}$$

Where $V_R$ (m$^3$) is the total released volume and $A$ (m$^2$) is the planimetric inundation area.

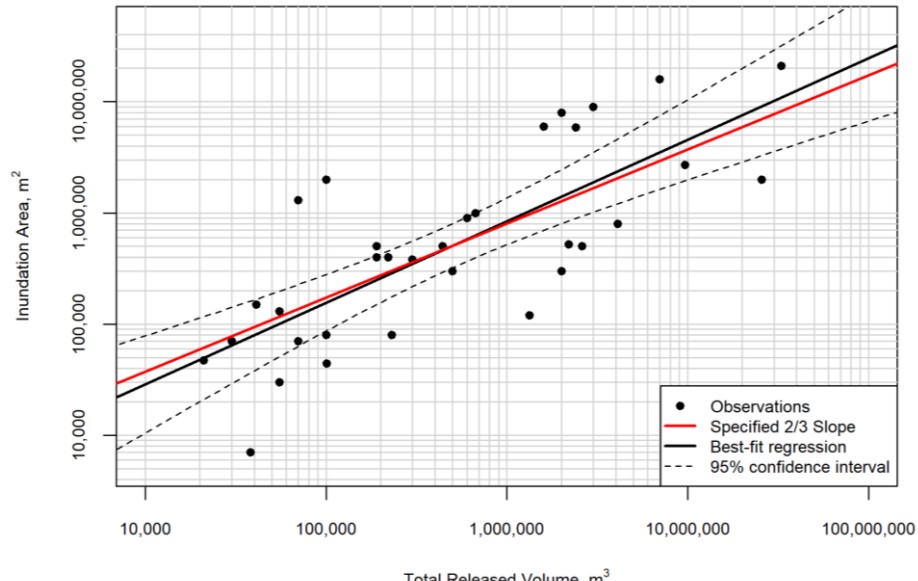

**Figure 6: Log-log scatter plot of planimetric Zone 1 inundation area versus total released volume for 33 tailings-flow cases (Table 3). The specified 2/3 slope regression line (in red) is fitted to the data. The best-fit regression line (in black) and the 95% confidence intervals (dashed lines) of the best-fit regression are plotted for comparison.**

**Table 4. Statistical results of the regression analysis for the best-fit and specified 2/3 slope models.**

| Parameter | Best-fit regression | Specified 2/3 slope |
|---|---|---|
| Slope ($\alpha$) | *0.73* | *0.67* |
| Intercept of line at log V =0 (Log($\beta$)) | *1.52* | *1.90* |
| $\beta$ | *33* | *80* |
| Number of data, n | *33* | *33* |
| Standard error of model, $\sigma$ | *0.56* | *0.55* |
| Standard error of volume coefficient | *0.11* | *NA* |
| Standard error of intercept | *0.65* | *0.10* |
| Coefficient of determination, $r^2$ | *0.58* | *0.57* |

The power law form of the equation: $A = (\beta) V^{\alpha}$; The linear form of the equation in log-log scale: $Log(A) = \alpha \, Log(V) + Log(\beta)$. For $\alpha = 2/3$, $\beta = c$ coefficient *in Eq. (1).*

The residuals (i.e. observed inundation area minus predicted inundation area) of the regression line with a specified 2/3 slope were analyzed to investigate if the variation could be explained through qualitative descriptions of the tailings type or confinement of the tailings runout path. This analysis was completed by plotting the distribution of the regression residuals as a box plot, where the lowest bar is the minimum of the residual distribution, the lower box represents the first quartile to the median residual, the upper box is from the median to the third quartile, and the upper bar is the maximum of the residual

distribution. If the distributions show stratification (e.g. one distribution has all four quartiles that are lower than the quartiles for a second predictor), it is an indication that there is a consistent difference in behaviour based on the descriptive predictors. Fig. 7a shows that, for a given volume, the inundation area for unconfined flow paths tends to be smaller than that for confined flow paths. Similarly, Fig. 7b shows that, for a given volume, the inundation area for hard rock mine tailings tends to be smaller than that for soft rock mine tailings. While these differences in the mean or median values can also be observed in the respective

box plots, the regression residuals are not strongly stratified overall. These qualitative factors were used as indicator variables to fit new regression models, but the associations were found to be too weak for application.

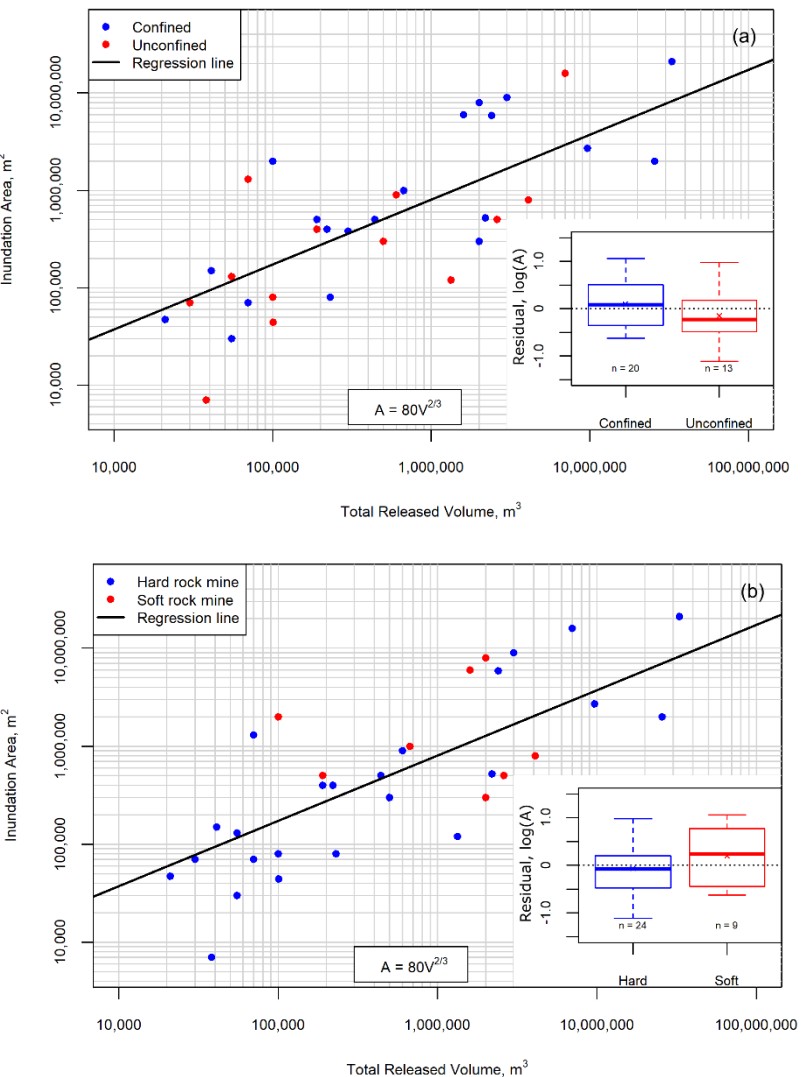

**Figure 7: Colour-coded data points with respect to path confinement (a) and tailings type (b). The solid black line is the specified 2/3**
**regression line. The insets show the box plots of the area-volume residuals for the bivariate regression line with a specified 2/3 slope.**

## 5 Discussion

The results listed in Table 4 indicate that Eq. (1) is a statistically-justifiable expression for the relationship between total released volume and planimetric Zone 1 inundation area, with coefficients of determination of 0.65 and 0.64 for the best-fit and the 2/3 slope regressions, respectively. Furthermore, the specified 2/3 slope line falls within the 95% confidence interval curves for the best fit regression, suggesting that the scaling relationship adopted by previous studies to characterize the geometry of other types of mass movements is also valid for tailings-flows. An analysis of the residuals from the regression grouped by tailings type and flow path confinement indicates that these factors have an effect on the mobility (i.e. the extent of planimetric inundation area for a given volume) of tailings-flows; soft rock mine tailings tend to have greater mobility than hard rock mine tailings, and confined flow paths tend to enhance mobility relative to unconfined paths, however, the data are not stratified enough to incorporate these factors into the regression analysis yet.

Figure 8 shows Zone 1 inundation area as a function of total released volume with the specified 2/3 regression line and its 95% prediction intervals, which account for the uncertainty of the individual data points. The difference between the lower and upper 95% prediction intervals reflects the variability of tailings-flows and the considerable uncertainties in the prediction of inundation area using this approach. Nonetheless, the prediction range that is achievable with this method is useful for first-order (screening level) risk assessment purposes, ideally within a probabilistic framework that acknowledges the level of uncertainty. This method is also useful for cross-checking numerical dam breach modelling results (i.e. to confirm that the simulated inundation area falls within a reasonable range relative to the cases included in this database). Note that, while the method is able to provide independent estimates of inundation area, it must be combined with other empirical and/or numerical methods that estimate cross-sectional area and runout distance in order to determine an appropriate spatial distribution of the estimated area, similar to the approaches that have been used for other hazard types, such as Iverson et al. (1998) and Mitchell et al. (2020). Further study is currently underway to estimate the cross-sectional area for tailings-flows and incorporate both volume-planimetric and cross-sectional area relationships in a GIS-based empirical model (Innis et al., 2020). Regardless of the approach used, significant professional judgement must be applied in interpreting the empirical results.

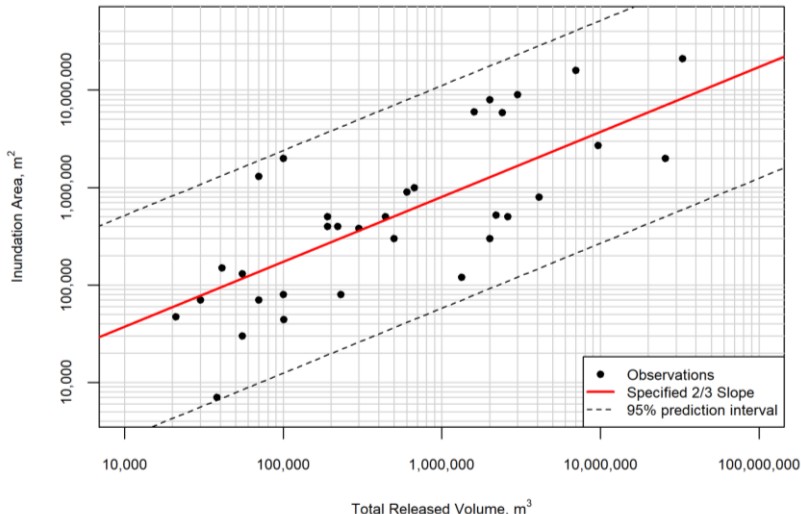

**Figure 8: Log-log scatter plot of planimetric Zone 1 inundation area versus total released volume for the 33 tailings-flow cases. The specified 2/3 slope regression line (in red) is fitted to the data and the 95% prediction intervals (dashed lines) of this regression line are also plotted.**

Figure 9 shows the area-volume scatter plot of tailings-flows alongside previously published data for lahars (Iverson et al., 1998), debris flows, rock avalanches (Griswold and Iverson, 2008), and mine waste dumps (Golder Associates Ltd., 1995)..
The tailings data points clearly show a positive linear pattern along with the other data, although the scatter is relatively high, especially at higher volumes. The area-volume data for tailings-flows show considerable overlap with other databases, corresponding with the upper volume range for debris flows, and the lower volume ranges for lahars and rock avalanches (Fig. 9). One of the possible impacts of the assumption that the released volume approximately matches the volume deposited downstream in Zone 1 (Section 3.2.1) is the deposited volume may be underestimated due to the entrainment of material along the flow path. This simplification may lead to overestimating the y-intercept of the regressions.

The differences between the *c* coefficient of Eq. (1) indicate the relative mobility of the various mass movement processes, on average (Berti and Simoni, 2007; Griswold and Iverson, 2008; Jakob, 2005). A comparison of *c* coefficients for different types of mass movements is shown in Table 2. The coefficient of $c = 80$ obtained for the tailings-flow data indicates that, on average, tailings-flows are less mobile than lahars but more mobile than mine waste dumps, debris flows, and rock avalanches for a given volume. There is a significant amount of scatter in all of the datasets shown in Fig. 9, which highlights the importance of considering the potential variability in these events for forward analysis (i.e. using probabilistic methods).

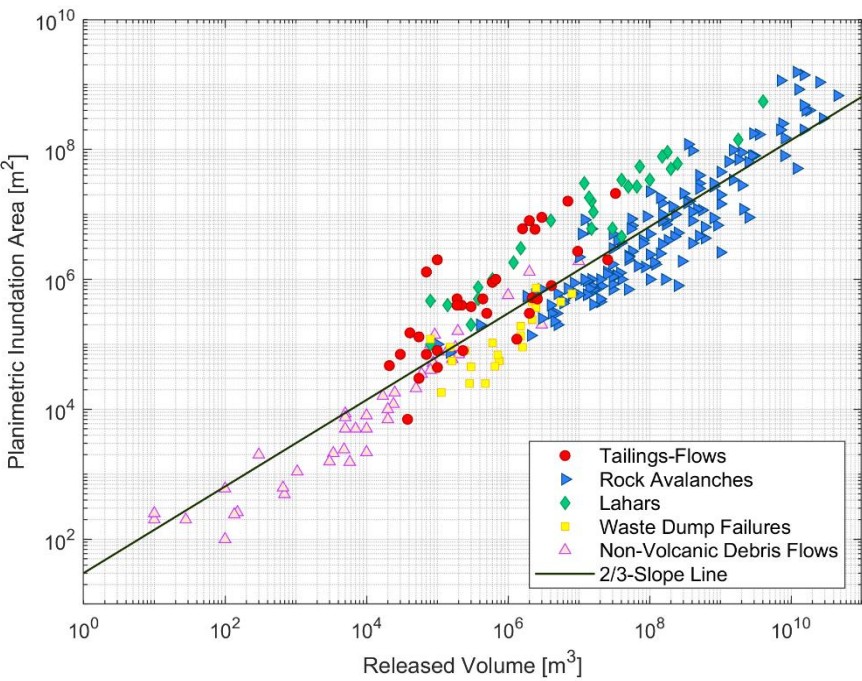

**Figure 9: Comparison of the runout inundation area as a function of flow volume for tailings-flows (red symbols; n = 33)), waste dump failures (yellow symbols; n = 22), lahars (green symbols; n = 27), non-volcanic debris flows (pink symbols; n = 44) and rock avalanches (blue symbols; n = 142). The black 2/3 slope line is drawn as a guide for visual comparison only.**

Five tailings dam breaches exhibit higher inundation areas than lahars for their given volumes (Fig. 9), and among those cases, the tailings dam breaches at the Ajka bauxite mine in Hungary in 2010 and the Mishor Rotem phosphate mine in Israel in 2017

(ID numbers 20 and 28 in Table 3) were examined in greater detail to demonstrate how site-specific information can be used to infer conditions that enhance mobility.

At the Ajka mine, a release of approximately 1.6 M $m^3$ of high-pH bauxite tailings, about 30% of which was solid residue, occurred through the northwest corner of the embankment (Bánvölgyi, 2018; Mecsi, 2013). The release produced a Zone 1 runout distance of approximately 18 km, despite the near-horizontal topography of the flow path (~0.2°), and covered

approximately 6 M $m^2$. The Ajka bauxite tailings had very weak geotechnical properties, with medium to high-plasticity, thixotropic (shear thinning) clays with very loose structure and slow consolidation rates, thus reducing pore fluid drainage and increasing the potential for liquefied flows (Mecsi, 2013). In addition to the volume of interstitial water, the bauxite tailings were overlain by a large supernatant pond that deepened towards the northwest corner of the impoundment; the average and maximum depth of the pond were 4.45 and 8 m, respectively, which greatly exceeded the maximum permitted pond depth of

1.5 m (Bánvölgyi, 2018) . We therefore attribute this secondary source of water, along with the observed thixotropic behaviour of bauxite tailings, to the augmented mobility of the Ajka tailings-flow.

The Mishor Rotem mine failure is estimated to have released approximately 0.1 M m$^3$ of highly acidic phosphogypsum tailings (Bowker, 2017). The ensuing tailings-flow travelled for 28 km through a dry creek channel with an average travel path angle of about 1.6° and inundated a Zone 1 area of approximately 1.8 km$^2$. As of yet, very limited information is available for this

tailings-flow, but a few authors have commented on the dominant contribution of high water content to the composition of phosphogypsum tailings (80-97%) compared to that of typical metal tailings (40-60%) (Bowker, 2017; Tao et al., 2010; Wang et al., 2014). We, therefore, propose three factors that contributed to the extreme runout behaviour (i.e. long runout distance and large inundation area for the given total released volume) of the Mishor Rotem tailings flow: i) high water content (interstitial and supernatant); ii) a narrow, dry channel situated within a stable desert environment with no physical obstacles to flow; and iii) a potential increase of the transported volume due to entrainment along the narrow channel.

Unlike natural hazards, tailings dams are human-made structures with impoundment volumes that increase over the course of mine operation. In most cases, when a dam breach occurs, only a portion of the impounded material is released. The amount of this portion depends on a variety of factors, such as the presence of a water pond, the tailings rheological properties, breach geometry, the age of the impounded material, and the triggering factors (Rico et al., 2008a).

The maximum volume that can be released in an extreme scenario equals the total impoundment volume. Compared with some types of landslides, the source volume of a tailings dam breach is relatively well-constrained. The uncertainty associated with this input parameter can, therefore, be accounted for explicitly when using Eq. (1) to make runout predictions. However, we note that relatively high confidence in the released volume estimate does not necessarily translate into high confidence in the inundation area estimate. Information on tailings type and topographic factors such as confined/unconfined travel path can

potentially be used to better constrain the uncertainty in predicting the inundation area as more data points are added.

Further investigation should focus on increasing the size of high-quality tailings-flow databases, which should lead to more robust statistical analyses. Some effort should also focus on quantifying the potential contribution of entrainment to the total volume of the deposited material.

## 6 Conclusions

Our empirical investigation of historical tailings dam breaches provides new insights into tailings-flow processes and characteristics and introduces new relationships that can potentially be used for first-order inundation mapping. In this study, we established a data compilation methodology and introduced a runout zone classification system to improve consistency and reduce uncertainties associated with previously reported data. Using this methodology, we compiled a database of 33 tailings dam breach case studies and estimated the planimetric Zone 1 inundation areas for all of the events. The degree of mobility of

the events in the database was investigated using a well-established semi-physical area-volume relationship, and the result was compared with similar relationships established for other mass flow processes. Our analysis suggests that the relationship $A = cV^{2/3}$ is a statistically valid relationship between total released volume ($V_R$) and planimetric inundation area ($A$). The $c$ coefficient of 80 from the analysis of our database suggests that, on average, tailings flows are less mobile than lahars ($c =$

200) but more mobile than mine waste dumps, debris flows ($c$ = 17-20), and rock avalanches ($c$ = 12-20). This paper is part of an ongoing project. We are currently building the database and investigating the effects of other attributes of the tailings and downstream topography, which could potentially be used to refine the area-volume empirical-statistical relationship.

## 7 Team list

- Negar Ghahramani, Ph.D. Candidate, Department of Earth, Ocean and Atmospheric Sciences, University of British Columbia, Vancouver, Canada. Email: nghahramani@eoas.ubc.ca
- Andrew Mitchell, Ph.D. Candidate, Department of Earth, Ocean and Atmospheric Sciences, University of British Columbia, Vancouver, Canada. Email: amitchell@eoas.ubc.ca
- Nahyan M. Rana, Ph.D. student, Department of Earth and Environmental Sciences, University of Waterloo, Waterloo, Canada. Email: nmrana@uwaterloo.ca
- Scott McDougall, Assistant Professor, Department of Earth, Ocean and Atmospheric Sciences, University of British Columbia, Vancouver, Canada. Email: smcdouga@eoas.ubc.ca
- Stephen G. Evans, Professor, Department of Earth and Environmental Sciences, University of Waterloo, Waterloo, Canada. Email: sgevans@uwaterloo.ca
- Andy Take, Professor, Department of Civil Engineering, Queen's University, Kingston, Canada. Email: andy.take@queensu.ca

## 8 Author contribution

Negar Ghahramani and Scott McDougall conceived the research idea and Negar Ghahramani developed the methodology. Negar Ghahramani and Nahyan M. Rana performed investigation. Andrew Mitchell and Negar Ghahramani performed the statistical analysis. Nahyan M. Rana, Negar Ghahramani and Andrew Mitchell verified the compiled database. Scott McDougall and Stephen. G. Evans and Andy Take Supervised the project. Scott McDougall and Stephen. G. Evans and Andy Take provided financial support for the project leading to this publication. Negar Ghahramani prepared the original draft with contributions from all co-authors.

## 9 Competing interests

The authors declare that they have no conflict of interest.

## 10 Acknowledgements

The authors would like to acknowledge the support of the Department of Natural Resources of Newfoundland for providing the information on the inundation area of the 2012 Gullbridge tailings dam breach. We would like to thank Sophia Zubrycky for her assistance with graphic design and Sahar Ghadirianniari for her assistance with data collection. The authors also wish to acknowledge valuable discussions with Vanessa Cuervo during the early stages of this work.

Funding:

This work was funded by a Fellowship (NG) from the University of British Columbia Department of Earth, Ocean and Atmospheric Sciences, as well as scholarships and grants from the Natural Sciences and Engineering Research Council of Canada (NSERC). This work is part of the CanBreach Project, which is supported by funding through an NSERC Collaborative Research Development Grant and funding from the following industrial partners: Imperial Oil Resources Inc., Suncor Energy Inc., BGC Engineering Inc., Golder Associates Ltd., and Klohn Crippen Berger.

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
