# Peer review of "Tailings-flow runout analysis: Examining the applicability of a semiphysical area-volume relationship using a novel database"

_Natural Hazards and Earth System Sciences, 2020_

## Referee Comment (RC1) · Renato Macciotta (Referee) · 4 Aug 2020

This is a very interesting manuscript that compiles an extensive literature review of the efforts made so far to develop hazard maps for tailings dam breaches. This is an important and timely research topic. Furthermore, a relationship between volume and inundation area is presented, that builds upon empirical work done by others for other geohazard phenomena. The manuscript is well written and I have only minor comments for the authors.

1.- Is it possible to include a metric for the goodness of fit between data and correlation in Table 2, as you did in Table 1; if available. Preference would be to use the same

metric (R-squared).

2.- Could you provide a quick explanation of how you calculate the 95% confidence interval in Figure 6 as opposed to Figure 8, early in the text? Is it to the data with respect to the area selected? Becomes somewhat confusing without an explanation as the data points clearly show that more half of them plot outside the boundary. Please review

---

## Author Comment (AC1) · 14 Aug 2020

The authors would like to thank Renato Macciotta (Referee) for the review of this manuscript and the provided comments.

In response to the first comment, we will add a new column to Table 2 including the R-squared values. Please see the attached edited Table 2.

In response to the second comment, when we are looking at the confidence interval on the regression we are looking at the uncertainty of the regression line itself, and not the individual data points (Figure 6). Likewise, when we plot the prediction interval

[Figure]

(Figure 8), then we are actually looking at the scatter in the data. Accordingly, the lines 222-223 and 262-264 of the manuscript will be modified as follows:

Line 222- Figure 6 shows the log-linear regression line for Zone 1 inundation area as a function of total released volume with the 95% confidence interval of the best-fit regression line. Please note that the 95% confidence intervals account for the uncertainty of the regression line and not the individual observations.

Line 262- Figure 8 shows Zone 1 inundation area as a function of total released volume with the specified 2/3 regression line and its 95% prediction intervals which account for the uncertainty of the individual data points. The difference between the lower and upper 95% prediction intervals reflects the variability of tailings-flows and the considerable uncertainties in the prediction of inundation area using this approach.
* * *
[Figure]

| Database Type | Equation | n | R-squared | References |
|---|---|---|---|---|
| Rock avalanches | $A = 76\ V^{0.57}$ | 76 | 0.78 | (Li, 1983)[a] |
| Rock avalanches | $A = 12\ V^{2/3}$ | 40 | - | (Hungr and Evans, 1993)[b] |
| Lahars | $A = 200\ V^{2/3}$ | 27 | 0.90 | (Iverson et al., 1998)[b] |
| Debris flows | $A = 17\ V^{2/3}$ | 90 | - | (Berti and Simoni, 2007)[b] |
| Debris flows | $A = 20\ V^{2/3}$ | 44 | 0.91 | (Griswold and Iverson, 2008)[b] |
| Rock avalanches | $A = 20\ V^{2/3}$ | 142 | 0.79 | (Griswold and Iverson, 2008)[b] |

[a] The original equation from (Li, 1983) is presented in power law format to facilitate comparison.
[b] $A$ and $V$ are planimetric area and flow volume, respectively ($A$ is in $m^2$ and $V$ is in $m^3$).

**Fig. 1.**

---

## Referee Comment (RC2) · Anonymous Referee #2 · 25 Sep 2020

General comments:

The authors present an empirical approach to constrain the relationship between tailings-flow volume and inundated area. Such approach is well known and the literature reports multiple application to different types of fast flowing landslides. Since tailings-flows have never been specifically addressed, the work done by the authors is appreciable and timely for publication. Results are relevant for the prediction of areas that can be impacted by tailings-flows. Although the method is simple and associated uncertainties relatively high, authors make a good job in recognizing limits and potential applicability of their results. I am not a native English speaker but I found it easy to

follow with clear and explanatory descriptions of the methods and results. The figures are well prepared with only minor flaws (see following) and numerical data are reported in tables that fosters the reproducibility of the method. In general, simple approaches towards the prediction of potentially catastrophic events are highly relevant and well within the scope of NHESS. In my opinion, the paper, in its present form, needs moderate modifications to further improve the quality of presentation before publication.

Specific comments:

line 97. Here you cite the dam factor parameter and in Table 1, the same is called predictor. I would stick to one definition and possibly describe the rationale behind this derived parameter. Furthermore, there is an erroneous under script parenthesis in the parameter equation.

lines 99-100. Unclear. Hf and dam factor are the same thing. Its relationship with runout distance improves with the updated database.

Table 2 (and Table 5). The dataset used by Berti and Simoni (2007) was later expanded with new cases (Simoni et al., 2011) resulting in a slightly different relationship: $A=18V^2/3$. 12. Simoni A., Mammoliti M., Berti M. (2011) Uncertainty of debris flow mobility relationships and its influence on the prediction of inundated areas. GEOMORPHOLOGY, 132: 249–259.

lines 183-184. The definition of uncertainty is incomplete. I guess it is the ratio (expressed as % in Table 3) between area of pixels intersected by the perimeter and total area of pixels mapping Zone 1. Please define unambiguously.

lines 209-210. Here you explain an important simplifying assumption. You should discuss this assumption and its possible impact on results. You can do it here or later when discussing the results (e.g., lines 275-280). In my opinion, the deposited volume is likely underestimated in your case due to entrainment of material along the flow path. Therefore, the Volume-Area relationship has higher intercept compared to

the method used by other researchers, which relates deposited volume and inundated area. However, I believe the assumption is reasonable because in case of tailings dams the release volume can be used of predictive purposes.

Figure 7. Please insert y-axis name and unit measure in the boxplots. Specify whether the regression line shown here is best-fit or 2/3 slope.

Figure 8. This figure contains the same info as Figure 6; only 95% prediction intervals are added. Consider adding them to figure 6 and eliminate Figure 8.

line 277. I could not find highlighted cases in Figure 9.

Figure 9. Please specify how your 2/3 slope fitting line is obtained in this case. Fonts used for this figure differ from other figures, please fix.

Table 5. Most of the data reported here have been reported in Table 2. Consider eliminating.

Discussion section. Here you describe a couple of interesting real cases in more detail. In my opinion, the paper would also benefit from the insertion of one (or more) example of predictions that could be obtained on your cases. More particularly, it would be interesting to compare on a map, the actual inundated area with the areas predicted using your equation and 95% prediction intervals.

line 314. The extreme runout behavior could have been also favored by an increase of the transported volume due to entrainment along the narrow channel that you describe.

---

## Author Response (AR1)

**Response to comments from the Referees**

**To Referee #1:**

The authors would like to thank Renato Macciotta (Referee) for the review of this manuscript and the provided comments.

1- Is it possible to include a metric for the goodness of fit between data and correlation in Table 2, as you did in Table 1; if available. Preference would be to use the same metric (R-squared).

In response to the first comment, we added a new column to the Table 2 including the R-squared values.

| Database Type   | Equation          | п   | R-squared | References                                |
|-----------------|-------------------|-----|-----------|-------------------------------------------|
| Rock avalanches | $A = 76 V^{0.57}$ | 76  | 0.78      | (Li, 1983) a                   |
| Rock avalanches | $A = 12 V^{2/3}$  | 40  | -         | $(Hungr and Evans, 1993)^b$               |
| Lahars          | $A = 200 V^{2/3}$ | 27  | 0.90      | $(Iverson \ et \ al., \ 1998)^b$          |
| Debris flows    | $A = 17 V^{2/3}$  | 90  | -         | (Berti and Simoni, 2007) b     |
| Debris flows    | $A = 20 V^{2/3}$  | 44  | 0.91      | (Griswold and Iverson, 2008) b |
| Rock avalanches | $A = 20 V^{2/3}$  | 142 | 0.79      | $(Griswold and Iverson, 2008)^b$          |

*a*The original equation from (Li, 1983) is presented in power law format to facilitate comparison. *b*A and V are planimetric area and flow volume, respectively (A is in  $m^2$  and V is in  $m^3$ ).

2- Could you provide a quick explanation of how you calculate the 95% confidence interval in Figure 6 as opposed to Figure 8, early in the text? Is it to the data with respect to the area selected? Becomes somewhat confusing without an explanation as the data points clearly show that more half of them plot outside the boundary. Please review.

In response to the second comment, when we are looking at the confidence interval on the regression we are looking at the uncertainty of the regression line itself, and not the individual data points (Figure 6). Likewise, when we plot the prediction interval (Figure 8), then we are actually looking at the scatter in the data. Accordingly, we will include more details in the text to make it more clear:

Line 224- Figure 6 shows the log-linear regression line for Zone 1 inundation area as a function of total released volume with the 95% confidence interval of the best-fit regression line. Please note that the 95% confidence intervals account for the uncertainty of the regression line and not the individual observations.

Line 266- Figure 8 shows Zone 1 inundation area as a function of total released volume with the specified 2/3 regression line and its 95% prediction intervals which account for the uncertainty of the individual data points. The difference between the lower and upper 95% prediction intervals reflects the variability of tailings-flows and the considerable uncertainties in the prediction of inundation area using this approach.

**To Referee #2:**

The authors wish to thank the anonymous referee for very detailed reviews of this manuscript.

Specific comments and the responses:

1- line 97. Here you cite the dam factor parameter and in Table 1, the same is called predictor. I would stick to one definition and possibly describe the rationale behind this derived parameter. Furthermore, there is an erroneous under script parenthesis in the

parameter equation.

lines 99-100. Unclear. Hf and dam factor are the same thing. Its relationship with runout distance improves with the updated database.

In response to the first and second comments, the dam factor and  $H_f$  are not the same. Larrauri and Lall (2018) presented a new predictor in their paper and called it  $H_f$  which is defined as  $H \times (V_{F/V_T}) \times V_F$ , while the dam factor is defined as  $H \times V_F$ . To make it more clear, we modified the manuscript as follows

*Lines* 99-100- *They introduced a new predictor, called*  $H_f$  *which is defined as*  $H \times (V_{F/V_T}) \times V_F$ , *where*  $V_T$  *is the total volume of the tailings impoundment and*  $V_F$  *is the total released volume.*

Table1- The word "predictor" is removed.

2- Table 2 (and Table 5). The dataset used by Berti and Simoni (2007) was later expanded with new cases (Simoni et al., 2011) resulting in a slightly different relationship:

A=18V2/3. 12. Simoni A., Mammoliti M., Berti M. (2011) Uncertainty of debris flow mobility relationships and its influence on the prediction of inundated areas. GEOMORPHOLOGY, 132: 249–259.

Thank you bringing this up, Table 2 has been updated to include the more recent information.

3- lines 183-184. The definition of uncertainty is incomplete. I guess it is the ratio (expressed as % in Table 3) between area of pixels intersected by the perimeter and total area of pixels mapping Zone 1. Please define unambiguously.

Thank you for the comment. The uncertainty values presented in Table 3 are the percentage uncertainty. We modified the definition of uncertainty in the text to be matched with percentage uncertainty values provided in Table 3. Here is the new definition:

Lines 185-186- The maximum percentage uncertainty due to image resolution was considered to be equal to the ratio of the total area of the pixels intersected by the perimeter of Zone 1 to the inundation area multiplied by one hundred.

4- lines 209-210. Here you explain an important simplifying assumption. You should discuss this assumption and its possible impact on results. You can do it here or later when discussing the results (e.g., lines 275-280). In my opinion, the deposited volume is likely underestimated in your case due to entrainment of material along the flow path. Therefore, the Volume-Area relationship has higher intercept compared to the method used by other researchers, which relates deposited volume and inundated area. However, I believe the assumption is reasonable because in case of tailings dams the release volume can be used of predictive purposes.

**This is a valid point. The following sentences are added to the manuscript:**

Lines 288-290- One of the possible impacts of the assumption that the released volume approximately matches the volume deposited downstream in Zone 1 (Section 3.2.1) is the deposited volume may be underestimated due to the entrainment of material along the flow path. This simplification may lead to overestimating the y-intercept of the regressions.

5- Figure 7. Please insert y-axis name and unit measure in the boxplots. Specify whether the regression line shown here is best-fit or 2/3 slope.

Thank you for the comment. Figures 7a and 7b are updated. The solid black line is the specified 2/3 regression line. This statement is added to the caption of Figure 7.

6- Figure 8. This figure contains the same info as Figure 6; only 95% prediction intervals are added. Consider adding them to figure 6 and eliminate Figure 8.

Thank you for the suggestion but we prefer to keep them separate.

In Figure 6, we plotted the 95% confidence interval of the best-fit regression to investigate the adaptability of the proposed relationship for tailings-flows. In Figure 8, which appears later in the Discussion section, we plotted the 95% prediction interval of the specified 2/3 slope to present the variability of tailings-flows and highlight the considerable uncertainties in the prediction of inundation area using this approach. Combining these intervals in one figure may result in misinterpretation and confusion.

7- line 277. I could not find highlighted cases in Figure 9. Figure 9. Please specify how your 2/3 slope fitting line is obtained in this case. Fonts used for this figure differ from other figures, please fix.

The sentence in Line 277 has been removed from the manuscript since we specified those two cases with their ID numbers later on in Section 5 (line 306).

In Figure 9, the 2/3 slope line is not a fit and it serves as a guide for visual comparison only. We modified the caption of Figure 9, accordingly.

8- Table 5. Most of the data reported here have been reported in Table 2. Consider eliminating.

The information in Table 5 is combined with Table 2 and Table 5 is removed.

9- Discussion section. Here you describe a couple of interesting real cases in more detail. In my opinion, the paper would also benefit from the insertion of one (or more) example of predictions that could be obtained on your cases. More particularly, it would be interesting to compare on a map, the actual inundated area with the areas predicted using your equation and 95% prediction intervals.

Thank you for the great suggestion. This study only presents the relationship between the planimetric inundation area and the release volume. This relationship must be combined with other empirical and/or numerical methods that estimate cross-sectional area and runout distance to determine an appropriate spatial distribution of the estimated area.

We are currently expanding this study by estimating the cross-sectional area for the tailings-flow cases that are presented in this manuscript and implementing both volume-planimetric and cross-sectional area relationships in Laharz, a GIS-based empirical model. The preliminary result is going to be published in the proceeding of the 2020 Tailings and Mine Waste Conference (Innis et al., 2020).

The Manuscript is updated as follows:

Lines 272-277- Note that, while the method is able to provide independent estimates of inundation area, it must be combined with other empirical and/or numerical methods that estimate cross-sectional area and runout distance in order to determine an appropriate spatial distribution of the estimated area, similar to the approaches that have been used for other hazard types, such as Iverson et al. (1998) and Mitchell et al. (2020). Further study is currently underway to estimate the cross-sectional area for tailings-flows and incorporate both volume-planimetric and cross-sectional area relationships in a GIS-based

empirical model (Innis et al., 2020): Regardless of the approach used, significant professional judgement must be applied in interpreting the empirical results.

Innis S., Ghahramani N., et al., (2020). "Automated Hazard Mapping of Tailings Storage Facility Failures". Tailings and Mine Waste 2020 (accepted).

10- line 314. The extreme runout behavior could have been also favored by an increase of the transported volume due to entrainment along the narrow channel that you describe.

Thank you for the input. We agree with your opinion. The new sentence is added to the manuscript:

[revised manuscript text omitted]